# Calpain Regulation and Dysregulation—Its Effects on the Intercalated Disk

**DOI:** 10.3390/ijms241411726

**Published:** 2023-07-21

**Authors:** Micah W. Yoder, Nathan T. Wright, Maegen A. Borzok

**Affiliations:** 1Biochemistry, Chemistry, Engineering, and Physics Department, Commonwealth University of Pennsylvania, 31 Academy St., Mansfield, PA 16933, USA; 2Department of Chemistry and Biochemistry, James Madison University, 901 Carrier Dr., Harrisonburg, VA 22807, USA

**Keywords:** calpain, calpastatin, intercalated disk, intercalated disc, heart, gap junction, desmosome, adherens junction

## Abstract

The intercalated disk is a cardiac specific structure composed of three main protein complexes—adherens junctions, desmosomes, and gap junctions—that work in concert to provide mechanical stability and electrical synchronization to the heart. Each substructure is regulated through a variety of mechanisms including proteolysis. Calpain proteases, a class of cysteine proteases dependent on calcium for activation, have recently emerged as important regulators of individual intercalated disk components. In this review, we will examine how calcium homeostasis regulates normal calpain function. We will also explore how calpains modulate gap junctions, desmosomes, and adherens junctions activity by targeting specific proteins, and describe the molecular mechanisms of how calpain dysregulation leads to structural and signaling defects within the heart. We will then examine how changes in calpain activity affects cardiomyocytes, and how such changes underlie various heart diseases.

## 1. Introduction

The heart beats 100,000 times per day [1]. During each contraction/relaxation cycle, cardiomyocytes exert up to 10 μN of force, subjecting these cells to orders of magnitude of more tensile strength than many other cells experience [2,3,4]. To accommodate these physical stressors, cardiomyocytes employ a complex proteinaceous structure, termed the intercalated disk (ICD), that both adheres the transverse ends of neighboring cardiomyocytes to each other and coordinates electrical and chemical impulses between these cells [5]. ICDs are only found in cardiac tissue, but the individual protein complexes—gap junctions, adherens junctions, and desmosomes (Figure 1)—are in all muscle and some non-muscle tissues [6,7,8]. Protein turnover, protein–protein interactions, and cell signaling of the various ICD components are tightly regulated. Conversely, when dysregulated, structures within the ICD alter cell signaling and tissue morphology, ultimately leading to a wide variety of heart-specific diseases [5]. In this review we will explore the specific ICD-related pathways influenced by the endogenous calcium-dependent protease calpain. We will then examine the clinical manifestations and cellular repercussions of altering these pathways.

## 2. Intercalated Disk Components, Functions, and Regulation

### 2.1. Gap Junctions

Gap junction complexes and other channels within the ICD modulate intercellular electrical and chemical signals to unify the myocardium’s rhythmic contractile beat [10]. Individual gap junction subunits, termed connexins (Cxs), multimerize into a hexameric membrane pore, which connects to another hexameric pore on the adjacent cardiomyocyte [11]. The entire structure creates a low resistance channel through which ions and molecules up to 1 kDa in size, such as inositol triphosphates and cAMP, can flow [11,12]. In the heart, the most common connexins—Cx40, Cx43, and Cx45—oligomerize in different combinations to modulate gap junction conductivity and activation [13]. Gap junction activation is complex, and can be stimulated by mechanical means, pH, calcium, voltage, oxidative stress, phosphorylation, and channel proteolysis on either side of the pore [14]. Each connexin is unique; Cx40 and Cx43 are 75% similar in the N-terminal 240 residues with no significant similarity in the C-terminal third of the protein [15,16,17]. The ratio of Cx40, Cx43, and Cx45 is tightly regulated, with the atria primarily expressing Cx43 and Cx40, the ventricles expressing mostly Cx43, and other tissues such as the SA and AV node expressing Cx45 [18]. Alterations in Cx expression are strongly correlated with cardiac disease [18].

While predominantly localized to the T-tubules, the L-type calcium channel (Ca_v_1.2) is also enriched at the intercalated disk near gap junctions [19]. This voltage-gated heteropentameric complex is primarily responsible for converting the change in membrane potential into an inward calcium flux in excitation–contraction coupling in the heart. While the L-type calcium channel is not physically part of the gap junction, it partially regulates connexin function; increased calcium levels through Ca_v_1.2 activation stimulates Cx43 activity [20,21]. Relatedly, Na_V_1.5 also localizes peripherally to gap junctions [22]. These inward sodium channels reside in small clefts between neighboring cardiomyocytes and help create a voltage gradient that triggers electrical excitation across the myocardial tissue, thus synchronizing myocardial contraction independent of the gap junction [10,11].

### 2.2. Adherens Junctions

The adherens junction (AJ) connects the non-sarcomeric F-actin networks of neighboring cardiomyocytes. The AJ inter-membrane region is comprised of N-cadherins, whose ectodomains associate with N-cadherins from the adjacent cardiomyocyte to dimerize in a catch-bond that can bear forces of around 35 pN [23]. A typical AJ contains around 2000 cadherins/µm^2^, which allows these protein clusters to resist tensile strength of up to at least 100 nN each [23,24,25]. Beneath the cell membrane, the intrinsically disordered cadherin tail binds to both p120 catenin and β-catenin, which in turn binds to α-catenin, which anchors to F-actin [7]. Functionally, AJs are the primary way that cardiomyocytes transfer force to the load-bearing cytoskeletal networks of neighboring cells [26]. To regulate this essential function, components of the AJ are in a constant state of protein turnover and can be strengthened or weakened as the biochemical or mechanical situation warrants. The regulation necessary for this kind of rapid response partially involves post-translational modifications; phosphorylation of the cadherin tail, p120, and β-catenin influence both protein–protein affinity and the recruitment of peripheral proteins to the AJ [27,28,29]. However, proteolysis of AJ proteins also plays a role in normal AJ maintenance and signaling. Specifically, proteolytic AJ fragments act as feedback mechanisms providing regulatory signals, and proteolyzed cadherin shedding is important in development [30].

### 2.3. Desmosomes

Desmosomes, the third protein complex of the ICD, connect the intermediate filament (IF) network of neighboring cardiomyocytes in a similar manner to how adherens junctions connect the actin cytoskeleton. Desmin, the predominant IF in the heart, is both stretch-resistant and significantly more flexible than actin filaments [31]. These physical attributes allow it to organize and corral the contents of a continually moving cell [32]. Other functions flow from this basal property of IFs being flexible and strong; IFs and their cellular connections regulate tissue durability, stability, and response to tension [32]. Given these functions, it is no surprise that mutations to IFs and to the components that connect IFs to other cellular structures are often implicated in cardiomyopathies [33]. At the desmosome, IFs connect to desmoplakin, which in turn binds to plakophilin and plakoglobin [8]. These two proteins bind to the unstructured cytoplasmic tail of the transmembrane cadherins desmoglein and desmocollin [8]. These cadherins hetero-dimerize extracellularly to cadherins from neighboring cardiomyocytes, thus forming the connection of the IFs of one cell to the IFs of its neighbor. Desmosomal proteins are controlled through genetic regulation, phosphorylation events which alter protein–protein interactions, and targeted proteolysis, initiated through both physical and biochemical upstream signals [34].

In the heart, these three complexes colocalize at the border between adjacent transverse cardiomyocytes, together forming the larger ICD. These structures exhibit significant cross-talk with each other; for instance, Na_V_1.5 co-precipitates with Cx43 and N-cadherin, and colocalizes with plakophilin [35]. In another example, loss of plakophilin at the desmosome results in gap junction remodeling [35]. Thus, the ICD is best viewed as an interconnected structure rather than a series of three independent structures that happen to be in close proximity [35]. For the sake of organization in this review, we will provide an overview of the calpain family of proteases and discuss how proteolytic cleavage regulates and dysregulates individual ICD components. We will then examine how these cleavage events lead to both normal and pathophysiology in the heart.

## 3. Calpain

### 3.1. Calpain Isoforms and Structural Organization

Calpain, or Ca^2+^-activated, neutral, and intracellular cysteine proteinase is present in most species, including lower organisms like bacteria and fungi and more advanced species including mammals [36]. There are 16 calpain gene products expressed in humans, with *CAPN1* and *CAPN2* being the most common (Table 1). These two gene products are expressed in all human tissues and are often referred to as μ-calpain and m-calpain, respectively, to denote their affinity for calcium. *CAPN5*, *CAPN7*, *CAPN10*, and *CAPN13–16* gene products are also expressed ubiquitously at lower levels [37,38]. The *CAPN16* gene product may be nonfunctional; it only encodes the N-terminal half of the protease core. Likewise, *CAPN4* does not encode a proteinase, but instead is another name for the small regulatory subunit of calpain [37,39]. The gene product of *CAPN3* is only expressed in skeletal muscle and that of *CAPN6* is predominantly found in embryonic muscles and the placenta. The *CAPN8* and *CAPN9* gene products dimerize to make the protease complex, termed G-calpain, found within the gastrointestinal tract. The gene products of *CAPN11* and *CAPN12* are expressed in the testis and the hair follicle, respectively [37,40,41]. While more than half a dozen calpains are expressed in the heart, most are expressed at low levels, have not been studied, or both. Therefore, this review will focus almost entirely on calpain-1 and -2, which are responsible for most of the calpain-dependent protein modifications at the ICD.

The canonical, functional calpain organization consists of two gene products; an 80 kDa catalytic subunit (*CAPN-X*) and a 30 kDa regulatory subunit (*CAPN4*; also called *CAPNS* for ‘small subunit’). The catalytic subunit has five domains: DI, DIIa, DIIb, DIII, and DIV, while the smaller regulatory subunit contains domains DV and DVI (Figure 2A). DIIa and DIIb together make up the central peptidase-C2-like fold. DIV, DV, and DVI, all of which contain multiple EF-hand motifs, are arranged on one side of DIIa/DIIb, while the active site and target binding cleft are on the opposite DIIa/DIIb face [41]. The active site, consisting of the catalytic triad cysteine, histidine, and a non-canonical asparagine, is broken up across DIIa and DIIb, with the active cysteine residue located in DIIa (human calpain-1 Cys115 and calpain-2 Cys105) and the histidine (human calpain-1 His272 and calpain-2 His262) and asparagine (human calpain-1 Asn296 and caplain-2 Asn286) residue in DIIb [42,43]. These two subdomains are connected via a flexible glycine–glycine motif [44]. The addition of calcium reorients DIIb relative to DIIa, moving the cysteine from 10.5 Angstroms away from the histidine to around 3.7 Angstroms (Figure 2B) [42]. The movement allows for an acid/base proton shuffling mechanism where a proton is transferred from the histidine to the asparagine and then from the cysteine to the histidine, creating a deprotonated cysteine nucleophile and thus activating the protease.

Calcium binds to calpain at numerous discrete locations, and each modulates calpain activity. Unlike many calcium-activated proteins, calpain’s main mode of action is not through large structural rearrangements of its many EF-hands. Instead, calpain participates in an ‘electrostatic switch’ mechanism. In the apo state, DIIb is tethered to DIII via electrostatic interactions between a lysine-rich helix in DIIb and the acidic activation loop in DIII (Figure 2C) [44,45,46]. Subsequent Ca^2+^ binding to this acidic loop neutralizes these attractive forces, freeing DIIb to ‘roll over’ towards DIIa, thereby forming a functional active site and target binding cleft [44]. The density of acidic residues on the DIII loop is likely the chemical basis for differential calpain calcium affinities; calpain-2 (m-calpain, exhibiting millimolar Ca^2+^ affinity) has fewer acidic residues in this loop compared to calpain-1 (μ-calpain, with micromolar Ca^2+^ affinity) [44,47]. Both DIIa and DIIb also bind calcium, which aids in the formation and stabilization of the target binding cleft [48,49]. In full-length calpain, DIIa is also tethered to the DI anchor helix, which is itself tightly bound to DIV, DV, and DVI (Figure 2D) [45,46]. This arrangement sterically constrains DIIa into a non-active conformation. Calcium binding to the DIV, DV, and DVI EF hands disrupts the DI interactions, thus releasing the DI anchor helix and promoting subsequent autoproteolysis at Ala9 and Gly19 (this and all further descriptions referenced to human calpain-2; accession code KAI4085066) [38,41,44,45]. In this final Ca^2+^-bound and DI-cleaved form, DIIa and DIIb are free to reorient, resulting in the active site residues being partially buried near the middle of a newly formed target binding cleft (Figure 2B).

Many proteases utilize deep binding pockets to achieve target specificity. Calpain does not. Instead, DIIa and DIIb create a shallow, long (>40 Å), convex target peptide binding cleft [49]. This cleft does not have an overabundance of any particular type of residue; instead, it contains a mixture of polar, charged, and hydrophobic amino acids [50]. Correspondingly, calpain targets exhibit low sequence homology. The length of the calpain binding cleft allows residues distal to the scissile bond to impact target affinity; calpain targets are enriched in small residues located 10–15 positions N-terminal to the cleavage site, as well as acidic side chains located 11–14 residues C-terminal to the cleavage site. Closer to the active site, calpain targets are enriched in leucine, valine, and threonine two residues before the scissile bond. The residue immediately after the cleavage site is usually small—often alanine or serine. Calpain targets often have proline amino acids two to five residues N-terminal to the scissile bond. Target sequences exhibit several other subtle trends, and these have been incorporated into online calculators that assess the likelihood that a given sequence will be cleaved by calpain (for instance: Deepcalpain, CalCleaveMKL, and GPS-CCD) [51,52,53]. An equally important criterion of determining calpain cleavage is secondary structure; long random coils are much better cleavage candidates than regions that adopt stable secondary structures, and solvent-exposed sequences are better targets than buried sites [54]. However, given the mutable characteristic of this cleavage site, experimental verification is the only way to confirm a putative target [55].

Calcium alters the calpain regulatory domains in a variety of subtle but functionally important ways. Both DIV and DVI are similar in structure, each containing five EF-hand motifs [42,44]. Four of the EF hands bind Ca^2+^ in the low to mid mM range [56], while the fifth EF-hand does not bind Ca^2+^ but instead aids in connecting subunits together [41]. Unlike canonical EF hands, most of those in calpain do not undergo a conformational change in the presence of calcium [41]. Rather, the EF-hands of these domains utilize changes in their electrostatic landscape to prevent improper calpain activation [41,44]. For example, Ca^2+^-binding in DIV EF2 positions a positive charge next to Lys7 in DI, thereby disrupting a number of DI-DVI salt bridges (Figure 2D) [41,44]. As another example, Ca^2+^ binding to EF3 in DIV results in the exchange of an electrostatic interaction for a hydrogen bond, which increases contact area between the DIIa/DIIb and regulatory domains by ~25% [41].

### 3.2. Calpain Activation and Regulation

Calpain is activated by several types of post-translational modifications and small molecules (Figure 3). Of these, the most obvious activator is calcium. Calcium binds calpain at numerous locations throughout the subunits; four in the DVI and DIV EF-hands, up to three in domain III, and two in domain IIa/IIb. The most functionally important are the binding events at DIII, which allows the formation of the proteolytic active site between DIIa and DIIb [44]. Calcium binding in domain II stabilizes the active site, although this is not necessary for activation. Intracellular calcium concentration of resting cardiomyocytes, in the mid-nanomolar range, is several orders of magnitude less than even the highest affinity calpain binding sites. The most likely explanation of this apparent concentration mismatch is that the local Ca^2+^ concentrations near Ca^2+^ channels, which includes the micro-environment at the ICD, can reach micromolar levels upon myocyte contraction. In an alternative mechanism, some calcium-binding proteins exhibit several-fold tighter calcium affinity in the presence of substrate. This is presumed to occur via a Monod–Wyman–Changeux (MWC)-like allosteric mechanism. In this mechanism, the apo- form of a Ca^2+^-binding protein is dynamic and occasionally samples target-binding conformations. When such orientations are briefly achieved, substrate binding can ‘lock’ the calcium-binding protein in a calcium-competent state, thus providing a pre-made calcium-binding site without the need for any additional protein conformational change or energy expenditure [57,58,59,60]. Neither of these hypotheses have been studied extensively in the calpain system and so remain merely speculative.

Calpains are also activated through proteolysis. This can be achieved either through autolysis of their own anchor helix to release DIIa from DI/DIV, or by other calpains performing this cleavage. Some calpains, such as calpain-3, seem to be truly autoproteolytic; the long nature of calpain-3 DI, which also contains both a nuclear translocation signal and a titin-binding site, allows it to interact with its own active site [61]. However, in calpain-1 and calpain-2 the anchor helix, DI, is ~4 nm from the proteolytic core, necessitating an intermolecular mechanism instead of auto-cleavage [42]. Calpain-1 can cleave calpain-2 at both the N-terminus and at the protease core, resulting in increased calpain-2 activation [42]. Interestingly, autolysis is not required for calpain activity, but is thought to amplify catalytic efficiency [42]. Cleavage at sites other than DI, such as those in DV, may reduce interactions between the large and small subunit, and thus, also further enhance calpain activity [62].

Calpains are positively regulated by phosphoinositides such as PIP_2_ and PIP [63,64]. These phospholipids bind to DIII and enhance calpain activity by both lowering the concentration of calcium required for autolysis and by increasing the V_max_ of calpain-1-driven proteolytic reactions [63,64,65].

Another activation method for calpains is through calmodulin-dependent protein kinase II (CaMKII) [66,67], suggesting β-adrenergic signaling may play a role in positive regulation [66,68]. In a positive feedback loop, CaMKII autophosphorylates and autoactivates in the presence of Ca^2+^-calmodulin, and this in turn stimulates the formation of a calpain–CaMKII complex that induces translocation of calpain to the cell membrane. In one example of how this affects cardiac health, blocking this translocation using a phospho-restrictive CaMKII variant inhibits calpain activity, which in turn preserves the function of the structural integrity protein α-fodrin and leads to improved myocardial membrane integrity [66]. Calpain-2 is also phosphorylated by ERK1/2 at Ser50, within the DIIa/DVI interface [69,70]. This activates calpain and leads to increased cell migration and adhesion [69]. Interestingly, this activation is calcium independent.

Just as calpains are activated through several mechanisms, they can also be inhibited by multiple means (Figure 3). Calcium, of course, can dissociate from calpain when intracellular calcium levels drop, allowing the DIIb/DIII interaction to re-engage. In another mechanism, activated calpain self-degrades with time, providing a built-in temporal inhibitory mechanism. In cell lines, calpain’s half-life is a relatively long 4–5 days [62]. In comparison, uncontrolled autolysis limits calpain half-life to the order of hours [71,72].

Calpain is strongly inhibited by the protein calpastatin. Upon binding to activated calpain, calpastatin transforms from a disordered protein into an elongated partial helical structure that wends through the target binding groove and occludes the calpain target binding region with low nanomolar affinity [43,73]. Thus, it acts as a competitive inhibitor. When unbound, or in low calcium levels, calpastatin aggregates in a kind of intracellular storage pool near the nucleus but becomes soluble after prolonged exposure to calcium [74]. Phosphorylation differentially modulates calpastatin activity; PKC-phosphorylated calpastatin almost completely abolishes the calpain–calpastatin interaction [75]. This is likely due to the addition of the negatively charged phosphate group at the C-terminus of calpastatin into a cluster of acidic residues at the bottom edge of the calpain binding site [75]. Calpastatin is also phosphorylated by PKA at a site outside of the calpain-interacting region, and this modulates both calpastatin solubility and cellular localization [75,76]. PKA also inhibits calpain directly. Phosphorylation of Ser369/Thr370, within the DIII/DIV interface, rigidifies the inactive conformation [70].

## 4. Calpain in Cardiac Pathophysiology

### 4.1. Normal Calpain Function

The calpain family of proteins promote many normal cellular behaviors related to migration, proliferation, and apoptosis. Specifically, calpain-2 localizes with and enhances the disassembly of focal adhesions through targeted cleavage events to promote cell motility [77]. Conversely, calpain inhibition in migrating cells is correlated with impaired rear cell retraction, increased tail length, and restrained cell movement [77]. Calpain also regulates normal cell cycle events. Calpain activation through a calpastatin degradation mechanism leads to vascular smooth muscle cell proliferation, and calpain overactivation contributes to tumor growth [78,79]. In some situations, calpain activation can also stimulate apoptotic cascades—in Ca^2+^-overloaded striated myocytes calpain cleaves apoptosis-inducing factor (AIF) from the inner mitochondrial membrane; cleaved AIF translocates into the nucleus and initiates DNA degradation [80]. In cardiomyocytes, calpain, activated through the TNF-α pathway, cleaves and activates poly-ADP ribose polymerase and caspase-3, ultimately leading to apoptosis [80]. Small molecule calpain inhibitors such as MDL-28170 tend to be cytotoxic with prolonged exposure, suggesting that calpain may be crucial in other yet-unknown basal cellular processes [81,82,83].

### 4.2. Calpain-Mediated Pathophysiology in Non-Cardiac Tissue—What Can This Teach Us about Calpain-Linked Function within the ICD?

Much of our understanding of calpain regulation and dysregulation comes from non-cardiac tissues. Moreover, although the ICD is exclusively in the heart, many cell types express the same or similar cell–cell junctional proteins. Calpain-mediated pathophysiology associated with these junctions across different tissues follows a common theme: calpain cleaves an integral junctional protein—either through calpain dysregulation or through a junction protein mutation, leading to increasing junctional fragility, and resulting in tissue-specific diseases that are often late to develop. The effects on cell function and disease types are dictated by the protein targeted and their tissue-specific function.

As an example of this general process, consider the molecular mechanism through which calpain exacerbates pulmonary edema. Fluid in the lungs increases the pressure on the surrounding endothelial cells, which in turn activates Piezo1 [84]. Piezo1 conducts calcium into the cell in direct response to the surrounding membrane flattening, and thus, chemically alters cells in response to physical cell stretch [85]. The resulting calcium influx activates both calpain-1 and calpain-2 in lung endothelial cells [84]. Once activated, the calpains cleave a host of AJ proteins including vascular-endothelial cadherins, β-catenin, and p120 catenin [84]. This increased proteolysis promotes more rapid AJ internalization and lysosomal degradation, resulting in a leakier endothelial lining and worsening the fluid buildup in the lungs [84].

Another example of calpain-mediated pathophysiology is found in eosinophilic esophagitis, a chronic allergic inflammatory disease characterized by esophageal eosinophilia (inflammation impairing contraction), epithelial hyperplasia (lesion/polyp formation), and dilated intercellular spaces all contributing to dysfunction of the esophagus. In the esophageal epithelial cells, aberrant calpain-14 activation cleaves native desmoglein-1, desmoplakin, and periplakin [86]. These cleavage events result in desmosomal breakdown, thereby compromising the epithelial architecture and barrier function through decreased cellular adhesion. This results in increased allergen penetrance and leads to an amplified inflammatory response [86].

A full description of calpain-mediated disease in non-cardiac tissue and their molecular pathways is outside of the scope of this review. Table 2 provides an overview of calpain-associated diseases that affect components of adherens junctions, gap junctions, and desmosomes.

### 4.3. Calpain-Mediated Pathophysiology in the Heart

#### 4.3.1. Calpain-Mediated ICD Targeting and Arrhythmogenic Cardiomyopathy

Arrhythmogenic Cardiomyopathy (ACM), historically Arrhythmogenic Right Ventricular Cardiomyopathy (ARVC), is a fatal inherited disease, often diagnosed post-mortem and characterized by progressive fibrofatty infiltration of the myocardium and a predisposition to sudden cardiac death [81]. Desmosomal disruption is linked to half of ACM cases [92]. ACM-linked desmosomal dysfunction is driven primarily but not entirely by genetic mutations. The best described example of non-genetic desmosomal-linked ACM is a study in which the authors examined the expression levels of wild-type desmoglein in ACM-afflicted myocardia [92]. In these samples, desmoglein was expressed in lower levels compared to non-ACM patients, despite having similar mRNA levels. This correlated with notably smaller desmosomes and increased inter-membranous space [92]. As desmoglein is a target of calpain, the authors suggest that the decrease in protein levels in ACM-afflicted myocardium is due to increased calpain targeting. Thus, it is conceivable that there are more ACM cases that are driven by the dysregulation of calpain or other proteases, but these are unreported due to the lack of an obvious genetic marker.

The highest percentage of ACM-related desmosomal mutations occur in plakophilin (25–50%), followed by smaller numbers in plakoglobin and desmoplakin [93]. A subset of these mutations is linked to calpain targeting. In one example, plakophilin p.Cys796Arg broadly destabilizes one of the armadillo-repeat regions, a short helical motif often involved in protein–protein interactions [93]. This leads to calpain hypersensitivity, resulting in protein cleavage and mislocalization away from the ICD to the area surrounding the nucleus [93].

In another example, select ACM-linked desmoplakin point mutations lead to calpain hypersensitivity. These clinical mutations (p.Arg451Gly, p.Ser299Arg, p.Ser507Phe, and p.Ser442Phe), which spatially reside in close proximity to each other within the N-terminal one-third of protein, expose a normally occluded calpain target site [94]. These mutations do not globally destabilize the desmoplakin structure [81]. Instead, they promote a localized unfolding event that predisposes these variants to calpain-dependent cleavage [81]. This calpain hypersensitivity was recapitulated in clinical samples, engineered heart tissue, and in in vitro studies [81]. Interestingly, calpain hypersensitivity was blocked not only through administration of calpain inhibitors like MDL-28170 but also by incorporating targeted corrective mutations into the desmoplakin variants that re-occlude the exposed calpain cleavage site [81,94]. This presents a potential path forward for treating calpain-dependent clinical diseases, where specific calpain target sites could be physically blocked using small molecules, biomolecules, or targeted mutations. Such an approach represents the first attempt to shift the scientific focus away from merely describing the etiology of calpain-linked ICD diseases to treating them.

#### 4.3.2. Calpain-Linked ICD Targeting and Atrial Fibrillation

Atrial Fibrillation (AF) is characterized by initial brief yet infrequent irregular and rapid heart rhythms, sliding into more chronic and sustained arrhythmia episodes lasting for several hours or longer [95]. AF can assume different clinical types depending on the timing of sinus rhythm resolution: paroxysmal AF resolves within 7 days while sustained AF can either last longer than 7 days (persistent) or indefinitely (permanent) [96,97]. AF builds cyclically to gradually further its presentation; AF-induced electrical remodeling slows down electrical conduction and shortens the effective refractory period in the atria, which in turn encourages prolonged AF episodes. Mechanistically, this is driven by fewer gap junctions in the atria myocardium [95]. Among other downstream effects, AF accompanied with a shortened refractory period results in an increased calcium ion influx and changes in phosphoinositide levels due to a higher rate of atrial myocyte activation [95,98].

The convergence of increased intracellular calcium levels, the perimembrane subcellular localization of calpains, and the fact that calpain activity increases in the presence of phospholipids suggests a potential link between AF and calpain hyperactivation. And, in fact, Cx43 is cleaved by calpain in mouse AF atria [95]. Likewise, in humans, Cx43 mRNA levels are unaltered in AF yet Cx43 protein levels (but not Cx40 or Cx45 levels) are significantly decreased, and calpain-1 levels are increased at the ICD in these same tissues [12]. In a canine AF model, calpain-1 activation correlates with decreased Cx40 levels, thereby categorizing Cx40 as another clinically important calpain substrate [99]. These data hint that calpain-dependent proteolytic cleavage of connexins is a molecular mechanism underlying AF variation.

Calpain-mediated proteolysis also influences L-type Ca^2+^ channel activity; calpain-induced premature degradation of these channels is associated with a reduced atrial effective refractory period in both paroxysmal and sustained AF, similar to what is seen with connexin degradation [12,100]. Etiologically, the reduction of L-type Ca^2+^ channels slow down the intercellular electrical conductance and the cellular response to electrical changes, which distorts Ca^2+^ homeostasis. This in turn results in higher intracellular calcium levels and increased calpain activity, which can then lead to even further L-type Ca^2+^ channel reduction.

The L-type Ca^2+^ channel also localizes to the T-tubule another membranous structure removed from the ICD. Here, calpain cleaves junctophilin-2, which in turn disrupts the precisely controlled spacing between plasma membrane the SR [101,102]. This spacial dysregulation disrupts the efficacy of the L-type Ca^2+^ channel in Ca^2+^-induced Ca^2+^ release in cardiomyocytes. This provides yet another example of how calpain indirectly affects cellular function.

#### 4.3.3. Calpain-Linked ICD Targeting and Myocardial Infarction

Blood vessel blockage to the heart, or myocardial infarctions (MIs), results in significant cardiac tissue death. Myocytes surrounding the infarct region are typically ischemic, i.e., in a hypoxic condition resulting from diminished blood flow to those myocytes. The ischemic event is associated with both elevated intracellular calcium levels and increased calpain activity [103]. MIs result in significant tissue reorganization, with matrix metalloproteases (MMPs) often being central to this reorganization due to their ability to degrade the extracellular matrix (ECM) [104]. Calpain indirectly activates several MMPs by cleaving the intermediate filament vimentin, which initiates MT1-MMP-dependent ECM reorganization [105]. In addition, calpain activates sphingosine-1-phosphate and T-antigen, which stimulates MMP transcription and MMP translocation to the cell membrane, respectively [106,107].

Once activated and localized to the cell membrane, MMPs target select ICD proteins. Calpain-activated MMP-7 cleaves Cx43, ablating Cx43’s ability to bind to the peripheral junctional scaffold protein ZO-1 [104]. This untethers the gap junction from the underlying cytoskeleton, leading to smaller gap junction size, altered gap junction distribution, and impaired electrical conductance [104]. Conversely, *MMP-7* gene deletion significantly improves post-MI survival, despite the loss of MMP-7-dependent tissue remodeling, and this is due to more robust intercellular electrical conductivity [104]. Other MMPs may work in similar calpain-dependent mechanisms [108,109], thereby indirectly implicating calpain in a worse prognosis following MI.

More directly, calpain-1 cleaves N-cadherin in post-MI cardiomyocytes due to the increased intracellular calcium levels [110]. As a result, cellular adhesion is disrupted and, post-MI, remodeling is aided. Cleavage of N-cadherin destabilizes the N-cadherin-Cx43 interaction, causing a mislocalization of gap junctions to areas outside the ICD [110]. Therefore, under select circumstances, increased calpain cleavage can result in further damage such as left ventricular dilation and corresponding cardiac dysfunction [110].

#### 4.3.4. Calpain-Linked ICD Targeting and Hyperhomocysteinemia

Hyperhomocysteinemia (HHcy) is a condition characterized by elevated levels of homocysteine and is associated with a high risk of cardiac injury and cardiomyopathy [111,112]. This condition initiates a signal transduction cascade that has many downstream results. One effect of initiating the signal cascade is translocation of calpain-1 into the mitochondria [111,112]. Mitochondrial Ca^2+^ levels broadly track with cytosolic levels, and under stress conditions can reach up mM levels [113]. Once inside the mitochondria, activated calpain-1 induces a burst of oxidative stress through an unknown mechanism, thereby activating mitochondrial-MMPs such as MMP-9. MMP-9 then translocates into the cytosol, where it can cleave proteins of gap junctions, adherens junctions, and desmosomes [112,114,115,116]. This provides yet another example where calpain indirectly disrupts ICD proteins resulting in decreased adhesion and conductance.

#### 4.3.5. Calpain-Linked ICD Targeting and Other Heart Dysfunctions

Some forms of coronary heart disease are linked to constitutive calpain-2 activation [15]. Recall, calpain-4 is the regulatory subunit of calpain, necessary for calpain-1 and calpain-2 activation. In a recent study, mice lacking calpain-4 and subjected to a prediabetic protocol had ameliorated heart dysfunction when compared to their wildtype counterparts [15]. Here, calpain represses angiogenesis and increases apoptosis in myocardial endothelial cells by promoting β-catenin proteolysis [15]. This disrupts the wnt/β-catenin signaling pathway, which regulates transcription of vascular endothelial growth factor, interleukin-8, and MMPs, all of which are major activators of angiogenesis [117].

### 4.4. Inhibiting Calpain in Heart Disease

From these descriptions, it would seem one obvious solution for many calpain-linked cardiac diseases would be to simply inhibit calpain with drugs such as MDL-28170 [36]. Such calpain-specific therapeutics do, in fact, show promise for heart-related diseases in either small doses or in short time frames [118,119]. However, since these drugs are not selective for overactive calpain and inhibit normal calpain activity, they face uncertainty as a long-term therapeutic for chronic diseases such as ACM, AF, and HHcy [81,82,83,118,119,120,121]. For more sustained and targeted inhibition, calpain activity may be able to be fine-tuned, for instance by blocking specific calpain targets or by targeting specific calpain activation mechanisms [66,67,94]. Further studies on how to modulate specific calpain activities and protect calpain targets are an obvious direction for future medical exploration.

## 5. Conclusions

The intercalated disk is a complex structure that bears the brunt of both physically and electrochemically coupling cardiomyocytes to each other. The individual protein complexes within the ICD work in concert to accomplish these tasks but are also functionally distinct. Proteins within each of these complexes—the gap junction, the adherens junction, and the desmosome—are subjected to calcium-dependent calpain degradation under specific pathophysiological conditions and, depending on which complex is targeted, lead to specific heart diseases. Both calpain dysregulation and calpain target mutations result in destructive calpain cleavage within the ICD. However, due to calpains’ other roles in promoting normal cardiomyocyte homeostasis, the long-term blockage of calpain activity is not a viable treatment option. Finding a way around this molecular roadblock represents the next frontier in calpain-dependent heart research. This is a still-emerging field; each year brings several new examples how calpain dysregulation of the ICD underlies heart disease. The continued discovery of these specific mechanisms contributes not only to increased recognition of calpain-related pathophysiologies, but also increases our understanding of how to manage and treat these conditions.

## Figures and Tables

**Figure 1 ijms-24-11726-f001:**
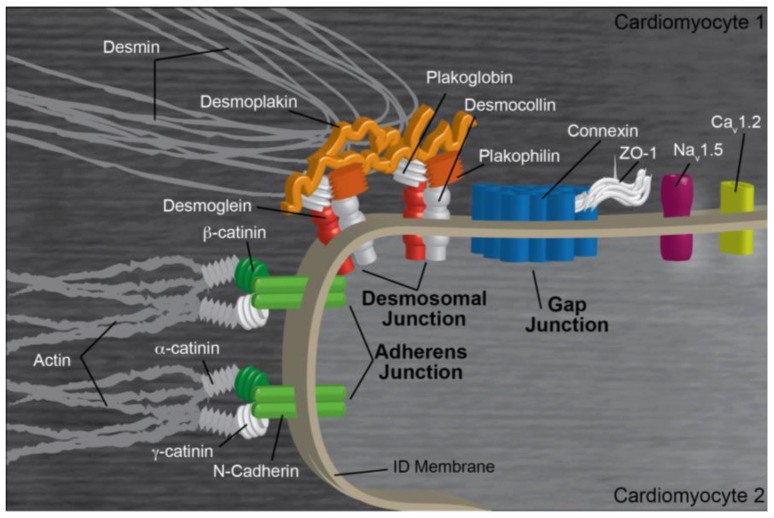
The intercalated disk linking two cardiomyocytes and major related proteins. The intercalated disk, depicted in beige, connects adjacent cardiomyocytes (light and dark grey) through three characteristic junctional complexes: gap junctions, adherens junctions, and desmosomal junctions. Proteins targeted by calpain are depicted in vibrant colors, while other components of the ICD are greyed. Image adopted from [9].

**Figure 2 ijms-24-11726-f002:**
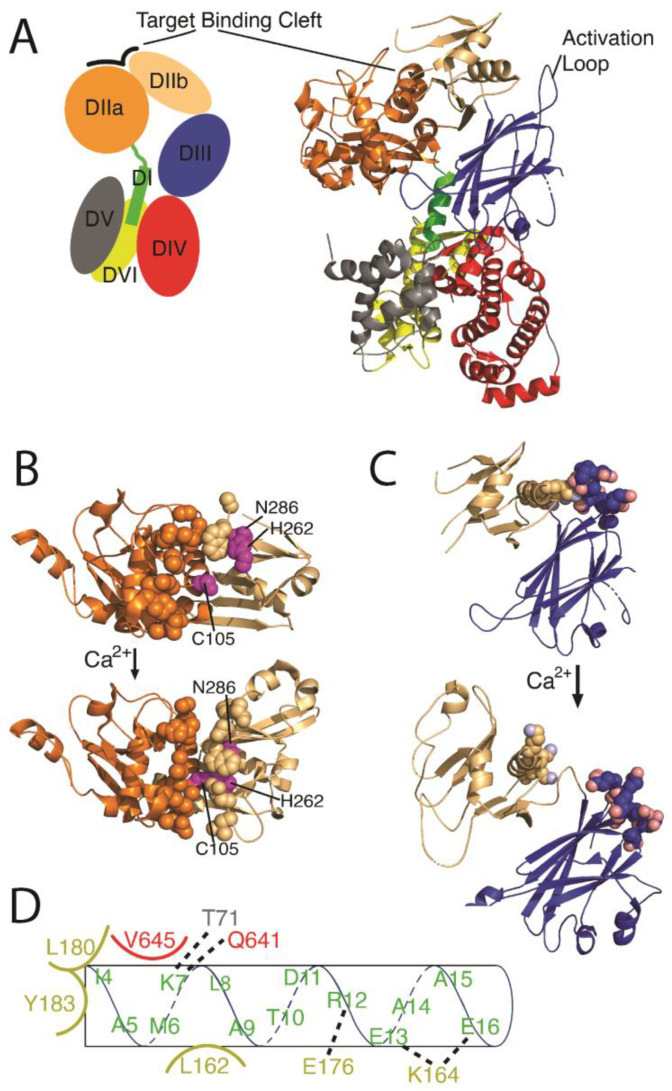
Calpain undergoes structural rearrangement in the presence of calcium. (**A**) Schematic of the entire apo m-calpain structure, including each domain orientation of both the large (DI-DIV) and small (DV-DVI) subunit. Adapted from pdb: 1u5i. (**B**) DIIa and DIIb reorient in the presence of calcium to form the active site and target binding cleft. Residues lining the target binding cleft in the Ca^2+^-bound state are depicted as spheres. The active site triad (Cys105 from DIIa, and His262 and Asn286 from DIIb) is in purple. The distance between Cys105 and His262 in the apo structure is 10.5 Å and decreases to 3.7 Å in the Ca^2+^ bound state. (**C**) Cartoon depicting the calcium switch mechanism between DIIb (beige) and DIII (blue). Acidic residues on the DIII activation loop (Glu392-Glu402) and lysines on the DIIb basic helix (Lys226, Lys230, and Lys234) are shown as spheres. (**D**) Schematic of hydrogen bond, electrostatic, and hydrophobic interactions between the sidechains of the DI anchor helix and DIV, DV, and DVI in apo-M-calpain. Ca^2+^ binding elsewhere in the structure disrupts these interactions, leading to a disordered DI, and eventually, auto-cleavage.

**Figure 3 ijms-24-11726-f003:**
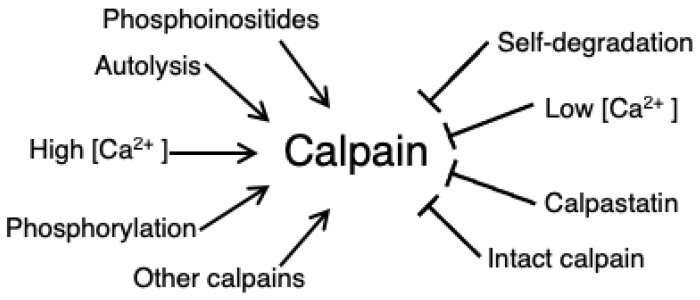
Calpain activators and repressors.

**Table 1 ijms-24-11726-t001:** Calpain Family of Proteins.

Gene	Protein	Tissue Expression	Notes
*CAPN1*	Calpain-1	Ubiquitous expression	Also known as μ-calpain
*CAPN2*	Calpain-2	Ubiquitous expression	Also known as m-calpain
*CAPN3*	Calpain-3	Expressed in skeletal muscle	
*CAPN4* or *CAPNS*	Calpain-4	Ubiquitous expression	Another name for the small regulatory subunit Also known as calpain-S
*CAPN5*	Calpain-5	Ubiquitous expression	Expressed in lower levels compared to calpain-1-2
*CAPN6*	Calpain-6	Expressed in embryonic tissue and the placenta	
*CAPN7*	Calpain-7	Ubiquitous expression	Expressed in lower levels compared to Calpain-1-2
*CAPN8*	Calpain-8	Expressed in gastrointestinal tissue	Complex with calpain-9 to make G-calpain
*CAPN9*	Calpain-9	Expressed in gastrointestinal tissue	Complex with calpain-8 to make G-calpain
*CAPN10*	Calpain-10	Ubiquitous expression	Expressed in lower levels compared to Calpain-1-2
*CAPN11*	Calpain-11	Expressed in testis tissue	
*CANP12*	Calpain-12	Expressed in hair follicle tissue	
*CAPN13*	Calpain-13	Ubiquitous expression	Expressed in lower levels compared to Calpain-1-2
*CAPN14*	Calpain-14	Ubiquitous expression	Expressed in lower levels compared to Calpain-1-2
*CAPN15*	Calpain-15	Ubiquitous expression	Expressed in lower levels compared to Calpain-1-2
*CAPN16*	Calpain-16	Ubiquitous expression	Expressed in lower levels compared to Calpain-1-2 Truncated protein, may be nonfunctional

Tabular representation of the calpain family of proteins noting tissue expression and unique notes regarding each family member.

**Table 2 ijms-24-11726-t002:** Calpain—Mediated Diseases in Non-Cardiac Tissue.

Disease	Dysregulation Mechanism	Calpain Target	Reference
Hyperpermeability Pulmonary Edema	Hyperactivity	VC-cadherin p120 catenin β-catenin	[84]
Breast Cancer Adenocarcinoma	Hyperactivity	E-cadherin	[87]
Alzheimer’s Disease/Bipolar Disorder/Neuronal Cancer	Hyperactivity	N-cadherin β -catenin	[88,89]
Neuronal Cell Death	Hyperactivity	Nav1.5	[90]
PLACK Syndrome	Hyperactivity	LOF mutations of calpastatin result in speculated proteolytic cleavage of desmosomal proteins	[91]
Eosinophilic Esophagitis	Hyperactivity	desmoglein, desmoplakin, periplakin	[86]

Tabular representation of calpain-mediated diseases that affect the adherens junction, desmosome, and gap junction in non-cardiac tissue.

## Data Availability

Data sharing not applicable.

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
