# Peer review of "Calpain Regulation and Dysregulation—Its Effects on the Intercalated Disk"

_ijms, 2023, doi:10.3390/ijms241411726_

Round 1

Reviewer 1 Report

This is very comprehensive review article which nicely reviews the progress in the field of research. The article was well prepared and easy to read. I do not see major concerns. 

In Chapter 3.2- Calpain activation and regulation, authors nicely summarized the regulation of calpain phosphorylation by CaMKII; however, other mechanisms are missing. For example, ERK1/2 phosphorylates and activates calpain as well.

In Chapter 4.3.2, calpain regulation of L-type calcium channel is mentioned. Since the L-type calcium channel is predominantly located to the T-tubules and junctophilin-2 is important in maintaining normal T-tubules, it would be nice to include calpain regulation of junctophilin-2 as well.

Author Response

In Chapter 3.2- Calpain activation and regulation, authors nicely summarized the regulation of calpain phosphorylation by CaMKII; however, other mechanisms are missing. For example, ERK1/2 phosphorylates and activates calpain as well.

ERK phosphorylation, which both activates and inhibits calpain, is now addressed (lines 277-279 and 298-299).

In Chapter 4.3.2, calpain regulation of L-type calcium channel is mentioned. Since the L-type calcium channel is predominantly located to the T-tubules and junctophilin-2 is important in maintaining normal T-tubules, it would be nice to include calpain regulation of junctophilin-2 as well.

A new paragraph on how calpain indirectly affects the L-type calcium channel at the t-tubule by cleaving junctophilin-2 and disrupts the spacing between membranes has been added (lines 422-427).

Reviewer 2 Report

In this review article, the authors reviewed calpain-linked intercalated disk regulation on the heart and calpain dysregulation leads to structural and signaling defects underlie various heart disease. The topics is interesting and updated for the average readers of the journal. Only several minor points I concerned:

1. In keywords showed both intercalated disk and intercalated disc. Please avoid repeating.

2. In Figure 2: Panel A, B, and C are not aligned. Please adjusted.

3. In title of Part 4: Calpain-Mediated Physiology, but the main content is calpain in pathophysiology, especially in heart diseases. Please modified.

Author Response

In keywords showed both intercalated disk and intercalated disc. Please avoid repeating.

The spelling for disk when referring to the intercalated disk is not a universal spelling. In fact, it can be spelled either disc or disk. Therefore, we have kept both versions in the key words to capture those searching regardless of the spelling.

In Figure 2: Panel A, B, and C are not aligned. Please adjusted.

Panel B has been moved down to align with Panel C. (p 6)

In title of Part 4: “Calpain-Mediated Physiology”, but the main content is “calpain in pathophysiology, especially in heart diseases”. Please modified.

This has been modified to read “Calpain in Cardiac Pathophysiology” (line 301).